# Microwave-Assisted Extraction Combined with In-Capillary [Fe(ferrozine)_3_]^2+^-CE-DAD to Screen Active Components with the Ability to Chelate Ferrous Ions from Flos Sophorae Immaturus (Flos Sophorae)

**DOI:** 10.3390/molecules24173052

**Published:** 2019-08-22

**Authors:** Tao Liu, Shanshan Wang, Huifen Ma, Hua Jin, Jin Li, Xuejing Yang, Xiumei Gao, Yanxu Chang

**Affiliations:** 1Tianjin State Key Laboratory of Modern Chinese Medicine, Tianjin University of Traditional Chinese Medicine, Tianjin 300193, China; 2Tianjin Key Laboratory of Phytochemistry and Pharmaceutical Analysis, Tianjin University of Traditional Chinese Medicine, Tianjin 300193, China; 3Chinese Medical College, Tianjin University of Traditional Chinese Medicine, Tianjin 300193, China; 4School of Pharmacy, Harbin University of Commerce, Harbin 150076, China

**Keywords:** capillary electrophoresis, MAE extraction, ability to chelate ferrous ions, Flos Sophorae Immaturus, Flos Sophorae

## Abstract

An efficient microwave-assisted extraction (MAE) combined with in-capillary [Fe(ferrozine)3]2+-capillary electrophoresis-Diode Array Detector (in-capillary [Fe(ferrozine)3]2+-CE-DAD) was developed to screen active components with the ability to chelate ferrous ions and determine the total antioxidant activity. The MAE conditions, including methanol concentration, extraction power, extraction time, and the ratio of material to liquid, were optimized by an L_9_(3^4^) orthogonal experiment. Background buffer, voltage, and cartridge temperature that affect the separation of six compounds were optimized. It was found that rutin and quercetin were the main components chelating ferrous ions in Flos Sophorae Immaturus (Flos Sophorae) by the in-capillary [Fe(ferrozine)_3_]^2+^-CE-DAD. The recoveries were ranged from 95.2% to 104%. It was concluded that the MAE combined with in-capillary [Fe(ferrozine)_3_]^2+^-CE-DAD method was a simple, reliable, and efficient tool for screening active components from the complex traditional Chinese medicine samples and evaluating their ability to chelate ferrous ions.

## 1. Introduction

As traditional Chinese medicine (TCM) is gradually adopted by the world, their effects have attracted more and more attention of researchers from various fields. Among those effects, antioxidant activity is important for many diseases. For example, some studies have shown that the anticancer activities of TCM components are directly or indirectly related to their antioxidant activity [1]. The antioxidant activity of TCM components is also closely related to many pharmacological effects, such as neuroprotective [2,3], antidepressant [4,5], anti-inflammatory [6], anticancer [7], hypoglycemic, and hypolipidemic [8]. Therefore, the research on antioxidant activity has become more and more important in TCM research.

As the strongest reducing agent in metal ions, ferrous ions are closely related to human aging and can produce strong oxidizing hydroxyl radicals and equivalent oxidizing power by reducing H_2_O_2_ [9]. The active components in TCM can chelate ferrous ions to form a stable complex and prevent ferrous ions from being oxidized or mediating other oxidation reactions. The formation of reactive oxygen species (ROS) and reactive nitrogen species (RNS) was mediated by ferrous ions chelated by some components of TCM [10,11,12]. It was found that tannic acid could block Fenton reaction by chelating ferrous ions to provide antioxidant activity, and the complex of tannic acid with Fe(II) would not react with H_2_O_2_ [13]. Therefore, the ferrous ions chelating ability of TCM is inextricably linked to its antioxidant activity. The antioxidant activity of TCM can be evaluated from the aspect of chelating ferrous ions [14].

Ultraviolet and visible (UV/VIS) spectrophotometry does not have enough specificity for determining the ability of components in the mixtures to chelate ferrous ions [15,16]. Moreover, the method possesses some disadvantages, including the consumption of more reagents and the complicated operation of metal-chelating assays. Recently, high-performance liquid chromatography coupled with post-column derivation (HPLC-PCD) was developed to determine the iron ions reduction ability of samples [17]. However, the post-column derivation made the operation more complicated. The capillary electrophoresis (CE) has the advantages of low solvent consumption and high efficiency, so it was widely applied in screening enzyme inhibitors, complex samples analysis, and derivatization [18,19,20] in recent years. At present, the in-capillary ABTS^+^-CE-DAD method [21] and DPPH-CE-DAD method [22,23] were established to evaluate the antioxidant activity of TCM. Based on these methods, an in-capillary [Fe(ferrozine)_3_]^2+^-CE-DAD was proposed to evaluate the ferrous ions chelating ability of TCM.

*Sophora japonica* is a medium-sized deciduous tree commonly found in China and many other countries. Flos Sophorae (FS) and Flos Sophorae Immaturus (FSI) are dry flowers and flower buds of Sophora japonica, respectively [24]. Many studies have shown that the antitumor, anti-inflammatory [25], antioxidant and radical scavenging [26], antibacterial [27], and other important activities of FSI and FS are related to their large amount of flavonoids. At present, the determination methods for the ability of FSI(FS) to chelate ferrous ions are mainly limited to UV spectrophotometry. Therefore, an in-capillary [Fe(ferrozine)_3_]^2+^-CE-DAD method was established for screening active components and determining the total ferrous ions chelating ability of FSI(FS).

The extraction methods of TCM make a great effect on its efficacy and activities. There are many extraction methods for TCM, including heating reflux [28], ultrasound [29], far-infrared [30], microwave [31], and ionic liquid-based pressurized liquid extraction (IL-PLE) [32], in which Microwave-assisted extraction (MAE) has the advantages of high extraction efficiency and is environment friendly. Furthermore, MAE is a reliable extraction method with high stability and reproducibility, which makes it superior to other traditional extraction methods [33]. Therefore, the MAE was combined with in-capillary analysis methods to evaluate the quality of FSI(FS).

Taking activity as an important aspect of quality control of traditional Chinese medicine is the trend of the development of TCM. To our knowledge, there is no literature on the application of in-capillary “[Fe(ferrozine)_3_]^2+^”-CE-DAD method for determining the ability of TCM to chelate ferrous ions. Therefore, we used the microwaves to assist in the extraction of FSI(FS) and in-capillary -CE-DAD method to evaluate the ability of extracts to chelate ferrous ions via screening the antioxidants and determining IC_50_ of the extracts. Factors affecting extraction and separation were optimized. The results of method validation proved that the microwave-assisted extraction (MAE) combined with in-capillary [Fe(ferrozine)_3_]^2+^-CE-DAD method was sensitive, reliable, highly efficient, and easy to operate and can be used for quality control of TCM.

## 2. Result and Discussion

### 2.1. Optimization of Microwave-Assisted Extraction Parameters

The processing and pretreatment of FSI(FS) have a great influence on its good activity and efficacy. To make the extract of FSI(FS) exert stronger ability of chelating ferrous ions, the factors affecting the microwave extraction of FSI(FS) were optimized by investigating the total inhibition rate. All extracts were diluted to a crude drug concentration of 5 mg·mL^−1^ to compare the total inhibition rate.

Based on the single factor test, the methanol concentration (A), microwave power (B), extraction time (C), and solid-liquid ratio (D) were selected as influencing factors. The levels of four factors are listed in Table 1), and L_9_(3^4^) was selected as an orthogonal test design to optimize extraction process parameters. All inhibition rates were calculated according to the Formula (1), in which $\bar{S}$ represents the peak area of [Fe(ferrozine)_3_]^2+^ when the sample extract was injected, and ${\bar{S}}_{\mathrm{blank}}$ represents the peak area of [Fe(ferrozine)_3_]^2+^ when deionized water was injected instead of the sample extract.

(1)$$\mathrm{Inhibition}\;\mathrm{rate}\;=\;\frac{{\bar{S}}_{\mathrm{blank}}-\bar{S}}{{\bar{S}}_{\mathrm{blank}}}\times 100\%$$

The results of the visual analysis are presented in Table 2, and the analysis of ANOVA are shown in Table 3. According to the software analysis, a significance level for the model (*p* < 0.05) was acquired. In Table 2, the K value was the average inhibition rate at each of the three levels of each factor. Therefore, the optimal level of each factor should be the factor level with the largest K value according to the largest donating rule. R-value was the range of K value (Kmax-Kmin).

Results of the visual analysis showed that the effects of methanol concentration, microwave power, extraction time, and solid-liquid ratio on the inhibition rate decreased successively, which proved that methanol concentration had the greatest influence on the inhibition rate of the sample, and the best factor level combination was A_1_B_3_C_3_D_1_. Analysis of variance showed that the methanol concentration, extraction time, microwave power had a significant effect on inhibition rate (*p* < 0.5). A_1_, B_3_, and C_3_ were, respectively, determined as the optimal factor levels. The optimum ratio of material to liquid was selected as 1:10 (g∙mL^−1^) with a less organic solvent. Thus, the optimal condition for microwave-assisted extraction of FSI(FS) was as follow: the extraction solvent was 50% (*v*/*v*) methanol, and the ratio of material to liquid was 1:10. The extraction was carried out for 7.5 min at a power of 750 w.

### 2.2. Optimization of On-Line Method

Background electrolyte (BGE) in the capillary can affect the migration and separation of the analytes. Thus, the factors, including pH, buffer salt concentration, additive concentration, voltage, and temperature associated with separation, were optimized. Each factor had an optimum value by investigating the migration time of six compounds (Figure 1) and resolution (Figure 2) of the analytes.

#### 2.2.1. Effect of Buffer pH

The effect of different pH values of Na_2_HPO_4_ buffer (6.5, 7, and 7.5), on the migration time and resolution of components, was investigated. The migration time of all analytes was decreased as the pH increased from 6.5 to 7.5 (Figure 1A). However, quercetin and [Fe(ferrozine)_3_]^2+^ had a maximum resolution, and kaempferol-3-rutinoside also had the best separation with narcissoside when pH value of the BGE was 7 (Figure 2A). Considering the migration time and resolution, the optimal pH value of BGE was chosen as 7.

#### 2.2.2. Effect of Buffer Concentration

The concentration of the Na_2_HPO_4_ had a great influence on the migration time and resolution. As can be seen from Figure 1B, the migration time of components increased significantly with the increasing buffer concentration. By comparing the resolution of kaempferol-3-rutinoside and narcissoside, the two components had the best separation when the concentration of Na_2_HPO_4_ was 50 mM. In summary, 50 mM with acceptable analysis time was selected as the optimal buffer concentration.

#### 2.2.3. Effect of SDS Concentration

Sodium dodecyl sulfate (SDS) is widely used in capillary electrophoresis analysis systems to modify background buffers to improve separation or increase sensitivity. In this study, the effect of SDS concentration (5 mM, 10 mM, and 15 mM) on migration time and resolution was investigated. As the concentration of SDS increased, the total migration time was slightly extended (Figure 1C). Quercetin and [Fe(ferrozine)_3_]^2+^ could achieve good separation at all concentrations of SDS (Figure 2C), and the resolution of kaempferol-3-rutinoside and narcissoside reached the maximum at 10 mM. Thus, 10 mM SDS with shorter total migration time was selected for further optimization.

#### 2.2.4. Effect of β-CD (β-cyclodextrin) Concentration

β-cyclodextrin (β-CD) has a good improvement in sensitivity and separation of analytes [34]. In this study, three concentrations (1 mM, 3 mM, and 5 mM) of β-CD were investigated. Although there was the shortest analysis time at 1mM (Figure 1D), the resolution of kaempferol-3-rutinoside and narcissoside was poorer (Figure 2D). Besides, the [Fe(ferrozine)_3_]^2+^ and other analytes had the best peak shape when the concentration of β-CD was 3 mM. Overall, 3mM was chosen as the optimal value of β-CD concentration.

#### 2.2.5. Effect of Acetonitrile Concentration

Acetonitrile has a focusing effect on the analytes, and it can improve the separation of hydrophobic and hydrophilic analytes [35]. The background buffer solution containing 0% to 4% acetonitrile had a smaller effect on the total migration time (Figure 1E), but the separation between kaempferol-3-rutinoside and narcissoside or quercetin and [Fe(ferrozine)_3_]^2+^ obtained an improvement at 2% (Figure 2E). Thus, 2% of acetonitrile was selected for the following study.

#### 2.2.6. Effect of Voltage and Temperature

Temperature and voltage also had a significant impact on the migration behavior of analytes. As shown in Figure 1F,G, the migration time of the analyte decreased significantly with the increasing voltage (18 kV, 20 kV, and 22 kV) and temperature (18 °C, 20 °C, and 22 °C). Although the migration time was slightly extended, the resolution shown in Figure 2F achieved the maximum value at 18 kV. Compared to the cassette temperature of 18 °C and 22 °C, the total migration time was shorter, and the separation of kaempferol-3-rutinoside and narcissoside was optimal at 20 °C. In summary, 18 kV and 20 °C were finally selected as separation voltage and cassette temperature, respectively.

It was concluded that the separation conditions for determining the ferrous ions chelating ability and screening the active components of FSI(FS) were as described below: 50 mM of Na_2_HPO_4_ buffer (pH = 7) containing 10 mM SDS, 3 mM β-CD, and 2% ACN (acetonitrile). The voltage and temperature were 18 kV and 20 °C, respectively. All analytes reached baseline separation, and the total analysis time was shortened to ten minutes. Therefore, the follow-up experiments were performed on this basis.

### 2.3. Method Validation

The calibration curves, linearity ranges, recoveries, limits of detection (LODs), and limits of quantification (LOQs) of rutin, narcissoside, quercetin, and kaempferol-3-rutinoside are summarized in Table 4. The degree of fit(R^2^) of the regressions equation ranging from 0.9978 to 0.9999 indicated good linearity. LODs and LOQs were in the range of 0.75 to 1.05 μg∙mL^−1^ and 2.50 to 3.50 μg∙mL^−1^, respectively. Recoveries of four components ranged from 95% to 104%, and all relative standard deviation (RSD) were lower than 5.1%, which proved that the extracted samples are stable and the method of microwave-assisted extraction was efficient.

The results of intra-day precisions, inter-day precisions, and stability for 24 h are summarized in Table 5. The remains of stability for 24 h ranged from 100% to 104%, and RSDs were lower than 3.51%, which indicated good stability of four components in the sample and [Fe(ferrozine)_3_]^2+^. Higher accuracy within 98–105% and RSD lower than 3% further proved that the established method was reliable and precise. Therefore, the method can be used to determine the ability of FSI(FS) to chelate ferrous ions.

### 2.4. Method Application

#### 2.4.1. Screening of Antioxidants of FSI(FS)

For the aim to screen antioxidants existing in FSI(FS), ferrous sulfate, sample extract, and ferrozine were injected sequentially into the CE-DAD system at 50 mbar pressure for several seconds. On comparing the electrophoretograms of online mixing H_2_O, sample extract and H_2_O with the electrophoretogram of online mixing ferrous sulfate, sample extract and ferrozine, it was found that the peak area of rutin and quercetin was significantly reduced, especially the peak of quercetin was almost absent, which indicated that both rutin and quercetin could chelate ferrous ions (Figure 3A). Under the optimal conditions, all components were identified to be rutin, kaempferol-3-rutinoside, narcissoside, quercetin by comparing the electrophoretogram of an extract with the electrophoretogram of mixed standard solution (Figure 4). Therefore, rutin and quercetin were two components chelating ferrous ions of FSI (FS).

#### 2.4.2. Determination of the Activity and Contents of the Herbal Samples

The developed method was applied to determine the total antioxidant activity and four flavonoids contents in three batches of FS and five batches of FSI (Table 6). To measure the total ferrous ions chelating ability of samples, 4 mM ferrous sulfate was reacted with the sample or equivalent volume of solvent in the capillary for 5 min and then mixed with 4 mM ferrozine. Comparing the two electrophoretograms in Figure 3B, the peak area of [Fe(ferrozine)_3_]^2+^ in the blue electrophoretogram was significantly reduced. The percentage reduction, i.e., the inhibition rate, was calculated from formula (a). The IC_50_ value was obtained by fitting the inhibition curve that measured the activity of herbal samples with six different crude concentrations. R-squared(R^2^) ranged from 0.9778 to 0.9984, IC_50_ values of FSI and FS were in the range of 16.25 to 18.41 mg·mL^−1^ and 22.78 to 25.44 mg·mL^−1^, respectively (Table 6). The citric acid has a strong ability to chelate ferrous ions [36]. Therefore, it was selected as a positive drug, and its IC_50_ value was 3.681 mg·mL^−1^.

It was found that there were some differences in the chelating ability of FSI(FS) from different places. It can be seen from Figure 5 that the total content of rutin and quercetin was closely related to the total activity of chelating ferrous ions. Therefore, rutin and quercetin were the main components chelating ferrous ions of FSI(FS), and they could be selected as active markers for evaluating ferrous ions chelating ability of FSI(FS). Through the determination of batches and positive drug, it was further demonstrated that the integrated activity of in-capillary [Fe(ferrozine)_3_]^2+^-CE-DAD method could be an effective technology for the quality control of TCM.

Many methods were applied to the extraction of FSI (Table 7). It was found that the extraction method adopted in this study had the advantages of less consumption of organic reagents, shorter extraction time, and higher extraction efficiency. The comparison with the existing methods for determining the ability to chelate ferrous ions is shown in Table 8. The method established in this paper integrated reaction and detection and achieved the purpose of screening active components of samples and measuring total antioxidant activity. Therefore, the method MAE combined with [Fe(ferrozine)_3_]^2+^-CE-DAD was efficient, simple, reliable, and can replace the traditional method to determine the ferrous chelating ability of the samples and screen the active components.

## 3. Materials and Methods

### 3.1. Chemicals and Reagents

The reference components, including rutin, kaempferol-3-rutinoside, narcissoside, and quercetin (more than 99% pure), were purchased from Chengdu Must Bio. Sci. and Tec. Co. Ltd. (Chengdu, China). The purity of the biological reagent ferrozine was 99%. Ferrous sulfate (FeSO_4_·7H_2_O) and citric acid purchased from Commio Chemical Testing Co. Ltd. (Tianjin, China). were of analytical grade. Deionized water was achieved from a Milli-Q system (Millipore, Milford, MA, USA). Acetonitrile (ACN) and methanol (MeOH) were bought from Merck (Chromatographic, Germany).

### 3.2. Preparation of Analytical Samples

All the standards and stock solution were prepared in methanol, and the mixed standard solution of each concentration was diluted by methanol. Ferrozine was dissolved in deionized water to 4 mmol·L^−1^ (mM) and stored at 4 °C. Ferrous sulfate was dissolved in deionized water to 4 mM before each use. The citric acid was dissolved in water to 4 mg·mL^−1^ and then diluted with water.

The extraction of samples was performed by Multiwave PRO SOLV (Anton Paar Co. Ltd., Graz, Austria). The operating condition was the mode of keeping power (no setting climb time) in which the temperature was optimized by adjusting the power. After 1 g of medicinal powder (through 80 mesh sieve) was extracted in 10 mL 50% (*v*/*v*) MeOH for 7.5 min at 750 w power, the weightlessness was compensated by adding the corresponding solvent. Subsequently, the mixture was cooled at room temperature and then centrifuged at 1000× *g* for 10 min. Finally, the supernatant was directly diluted with the corresponding extraction solvent to a crude drug concentration of 5 mg·mL^−1^ and kept at room temperature before analysis.

### 3.3. Instruments and CE Condition

All samples were analyzed by an Agilent 7100 capillary electrophoresis (CE) system (Waldbronn, Germany) with a Diode Array Detector. All analytes were separated on a fused silica capillary with a diameter of 50 µm and a total length of 50.2 cm (effective length of 42.2 cm) (Ruifeng, Handan, Hebei, China). Before the first use, the new capillary was activated by sequentially flushing with 1 M NaOH, 0.1 M NaOH, and deionized water. Before the samples were injected, 0.1 M NaOH for 1 min, deionized water for 1 min, and the background electrolyte (BGE) for 2 min were used to rinse the capillary in sequence. All samples were injected under 50 mbar. After the last analysis of one day, the analytical system was washed with 0.1 M NaOH and deionized water for 10 min. BGE needs to be updated every three runs to ensure that the method can be reproduced.

### 3.4. The Conditions and Principle of Method

The conditions, including pH of running buffer; the concentration of disodium phosphate, sodium dodecyl sulfate (SDS), β-cyclodextrin (β-CD), and ACN; the separation voltage, and cartridge temperature, were optimized. According to the principle of the traditional ferrozine method [35,37], after 4 mM ferrous sulfate was injected into CE system for 3 s, the samples extracted by microwave were injected for 5 s, incubated for 300 s, then 4 mM ferrozine was injected for 2 s, and the analytes were separated. In this process, ferrozine had a strong ability to chelate ferrous ions and was commonly used as an indicator for determining the ability of TCM to chelate ferrous ions. The active components of FSI(FS) would compete with ferrozine for chelating ferrous ions in the capillary. By comparing with the corresponding electropherogram of blank, the peak area of the active components or the chelate [Fe(ferrozine)_3_]^2+^ would be reduced. All of the six compounds had strong absorption at 260 nm, and [Fe(ferrozine)_3_]^2+^ also had absorption at 562 nm, so the detection wavelength was 260 nm.

### 3.5. Method Validation

Based on optimized extraction and separation conditions, active components were screened by comparing the electrophoretograms of in-capillary mixed ferrous sulfate, sample extracts, ferrozine with the electrophoretograms of in-capillary mixed sample extracts, and deionized water.

To verify the feasibility of the method, the calibration curves were fitted by injecting six concentrations of the mixed standard solution. The limit of detection (LOD) and the limit of quantification (LOQ) were obtained by electropherograms of the standard solution. A known concentration of the mixed standard solution was added into the extraction tank and then extracted together with the crude drug to obtain recoveries. On this basis, low, medium, and high concentration of mixed standard solution were used for evaluating the precision and stability for 24 h. Stability for 24 h was determined at six timepoints in one day. The accuracy of precision in one day or three days and remains of stability was the ratio of the measured concentration to the actual concentration.

## 4. Conclusions

The microwave-assisted extraction combined with in-capillary [Fe(ferrozine)_3_]^2+^-CE-DAD method was successfully established to screen active components and evaluate the ability of traditional Chinese medicine to chelate ferrous ions. The microwave extraction adopted in this study can be considered as an eco-friendly, green, and fast technique for the extract preparation because of less consumption of organic reagents, shorter extraction time, and higher extraction efficiency. Besides, it contributed to enhance the ability of the FSI and FS extracts to chelate ferrous ions. Rutin and Quercetin were screened as the target compounds to chelate ferrous ions in Flos Sophorae Immaturus and Flos Sophorae. Compared with the traditional methods, the online method of integrating reaction and detection not only extremely simplified the whole analysis procedure but also increased the sensitivity and reduced the consumption of reagent. Moreover, in-capillary screening of active components from TCM provides a comprehensive evaluation for the chelation of ferrous ions. Consequently, the proposed MAE extraction combined with in-capillary [Fe(ferrozine)_3_]^2+^-CE-DAD method can be a practical and effective technology for the evaluation of the activity of TCM. Furthermore, it can be used for quality control of TCM and can provide certain method strategies for industrial applications.

## Figures and Tables

**Figure 1 molecules-24-03052-f001:**
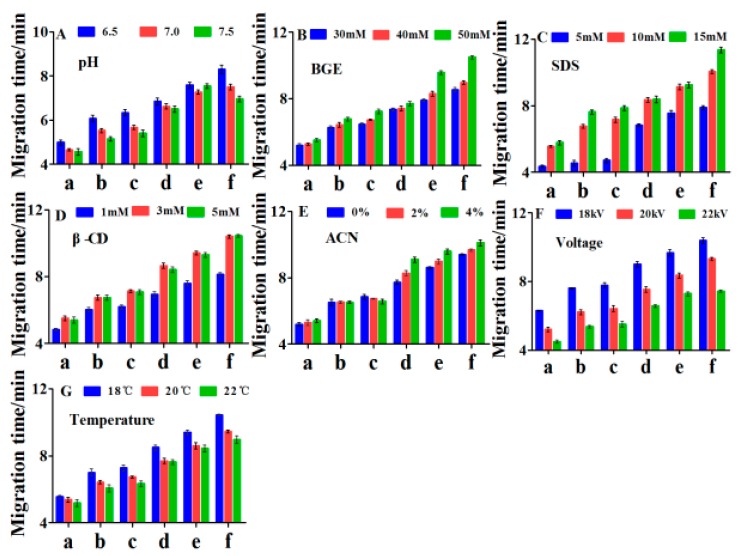
The effects of parameters on migration time of six peaks. (**A**) The effect of buffer pH on migration time, (**B**) the effect of buffer concentration on migration time, (**C**) the effect of SDS concentration on migration time, (**D**) the effect of β-CD concentration on migration time, (**E**) the effect of acetonitrile concentration on migration time, (**F**) the effect of voltage on migration time, (**G**) the effect of temperature on migration time.(a: Rutin, b: Kaempferol-3-rutinoside, c: Narcissoside, d: Ferrozine, e: [Fe(ferrozine)3] 2+, f: Quercetin). BGE: background electrolyte; SDS: sodium dodecyl sulfate; β-CD: β-cyclodextrin; ACN: acetonitrile.

**Figure 2 molecules-24-03052-f002:**
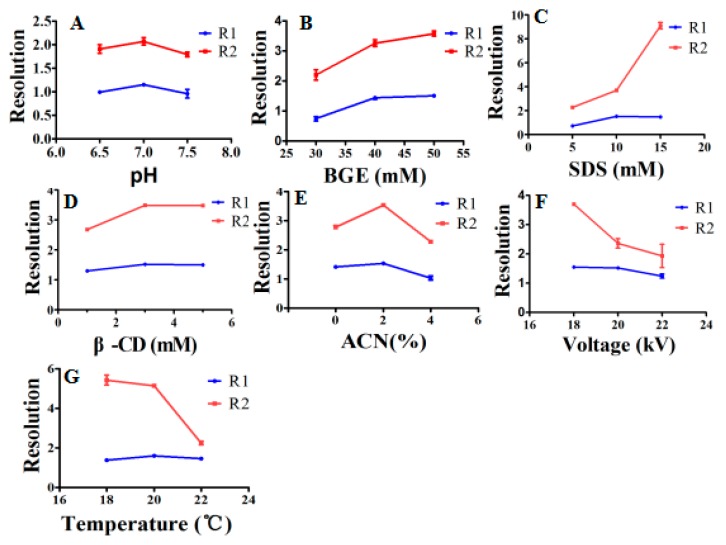
The effects of parameters on R1 and R2.;(**A**) The effect of buffer pH on R1 and R2, (**B**) the effect of buffer concentration on R1 and R2, (**C**) the effect of SDS concentration on R1 and R2, (**D**) the effect of β-CD concentration on R1 and R2, (**E**) the effect of acetonitrile concentration on R1 and R2, (**F**) the effect of voltage on R1 and R2, (**G**) the effect of temperature on R1 and R2.. (R1: The resolution of kaempferol-3-rutinoside and narcissoside; R2: The resolution of [Fe(ferrozine)_3_]^2+^ and quercetin).

**Figure 3 molecules-24-03052-f003:**
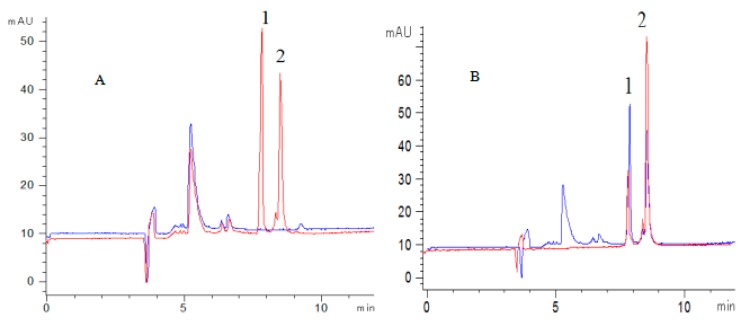
The typical electrophoretogram (**A**) The electrophoretogram of online mixing sample extract with (red) or without (blue) ferrous sulfate and ferrozine (*n* = 3). (**B**) The electrophoretogram of online mixing ferrous sulfate, ferrozine with (blue) or without (red) sample extract (*n* = 3). (1 Rutin, 2 Kaempferol-3-rutinoside, 3 Narcissoside, 4 Ferrozine, 5 [Fe(ferrozine)_3_] ^2+^, 6 Quercetin).

**Figure 4 molecules-24-03052-f004:**
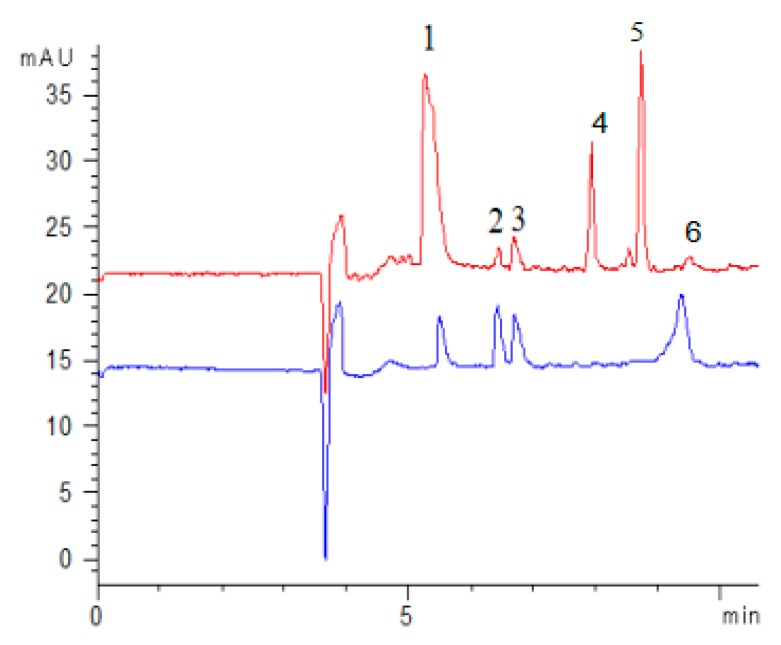
Capillary electropherograms of four compounds in mixed standard solution (red) and sample extracts (blue) (*n* = 3), (1 Rutin, 2 Kaempferol-3-rutinoside, 3 Narcissoside, 4 Ferrozine, 5 [Fe(ferrozine)_3_] ^2+^,6 Quercetin).

**Figure 5 molecules-24-03052-f005:**
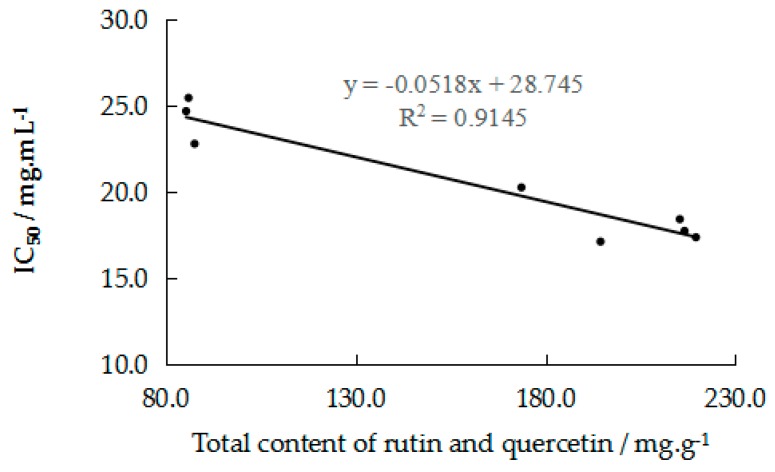
Relationship of the total contents of rutin and quercetin and IC_50_ of samples.

**Table 1 molecules-24-03052-t001:** Factors and levels of orthogonal test.

Factors	Levels		
	1	2	3
(A) Methanol concentration (%)	50	75	100
(B) Microwave time (min)	2.5	5	7.5
(C) Microwave power (W)	250	500	750
(D) Solid/liquid ratio (g∙mL^−^^1^)	1:10	1:20	1:50

**Table 2 molecules-24-03052-t002:** Arrangement and results of L_9_ (3^4^) orthogonal test.

No.	Factors				Inhibition Rates (%)
	A	B	C	D	
1	1	1	1	1	31.12
2	1	2	2	2	40.28
3	1	3	3	3	52.69
4	2	1	2	3	31.02
5	2	2	3	1	46.16
6	2	3	1	2	34.48
7	3	1	3	2	28.81
8	3	2	1	3	15.73
9	3	3	2	1	26.4
K_1j_ ^a^	41.363	30.317	27.110	34.560	
K_2j_	37.220	34.057	32.567	34.523	
K_3j_	23.647	37.857	42.553	33.147	
R_i_^b^	17.716	7.540	15.443	1.413	
O ^c^	A_1_	B_3_	C_3_	D_1_	

Note: a: Logesh Kij = (1/3) ∑mean inhibition rates of samples at factor j, j means extraction factor and i means setting level here, respectively. (j = A, B, C, D). b: Logesh Rij = max {Kij} − min {Kij}. c: O means the optimum condition.

**Table 3 molecules-24-03052-t003:** Analysis of variance (ANOVA) for the orthogonal test.

Source	Sum of Squares	Degrees of Freedom	*F*-Value	*p*-Value
A	515.283	2	132.327	*
B	85.279	2	21.900	*
C	368.005	2	94.506	*
D	3.894	2	1	
Error	3.89			

Note: * Indicates a significant impact.

**Table 4 molecules-24-03052-t004:** The calibration curves, linearity ranges, LODs (limits of detection), LOQs (limits of quantification), and recoveries of four compounds (*n* = 6).

Compounds	Regression Equation	r	Linearity Range (μg·mL^−1^)	LOD (μg·mL^−1^)	LOQ (μg·mL^−1^)	Recovery	
						Average (%)	RSD (%)
Rutin	Y = 0.2922x − 0.2637	0.9999	62.5–2000	1.05	3.50	104	4.33
Kaempferol- 3-rutinoside	Y = 0.2882x + 0.225	0.9988	3.125–100	0.9	3	96.3	1.10
Narcissoside	Y = 0.3255x − 0.5841	0.9978	5–160	1.05	3.50	102	4.63
Quercetin	Y = 0.8004x − 0.6256	0.9995	3.125–100	0.75	2.50	95.2	5.10

Note: RSD means relative standard deviation.

**Table 5 molecules-24-03052-t005:** Intra-day and Inter-day accuracy and precision, the stability of compounds. Ca is the actual concentration (*n* = 6).

Compounds	Ca (µg∙mL^−^^1^)	Intraday		Interday		Stability for 24 h	
		Accuracy (%)	RSD (%)	Accuracy (%)	RSD (%)	Remains (%)	RSD (%)
Rutin	250	98	2	99.4	1.75	102	1.01
	500	101	0.873	100	0.853	102	3.51
	1000	100	0.299	99.2	1.39	101	0.73
Kaempferol-3-rutinoside	12.5	101	1.99	101	2.30	104	1.91
	25	103	2.98	102	2.39	101	2.37
	50	101	1	100	1.03	102	3
Narcissoside	20	104	0.951	104	2.32	103	2.32
	40	102	1.14	101	1.27	103	1.49
	80	101	0.580	101	0.735	101	2.09
Quercetin	12.5	102	1.56	101	1.63	100	1.65
	25	101	1.78	100	1.22	100	1.54
	50	99.9	0.830	100	1.15	100	1.44
[Fe(ferrozine)_3_]^2+^	−	−	0.768	−	0.580	−	0.908

Note: − means be not calculated.

**Table 6 molecules-24-03052-t006:** Contents of four compounds in different samples (mg/g), and the IC_50_ values of samples and Citric acid (*n* = 3).

Samples	Flos Sophorae Immaturus					Flos Sophorae			Citric Acid
	HeNan	HeBei	GuangXi	ShanDong	ShanXi	HeBei	HeNan	ShanDong	
Rutin	210.46 ± 3.49	208.81 ± 1.64	211.48 ± 0.79	188.11 ± 5.71	164.91 ± 5.24	78.09 ± 1.13	79.45 ± 1.11	76.91 ± 2.66	-
Kaempferol-3-rutinoside	10.73 ± 0.48	10.51 ± 0.66	14.65 ± 1.31	7.78 ± 0.6	11.36 ± 0.69	6.98 ± 0.66	5.65 ± 0.16	6.8 ± 0.53	-
Narcissoside	6.69 ± 0.32	3.9 ± 0.39	8.22 ± 0.8	6.41 ± 0.46	4.83 ± 0.54	2.24 ± 0.29	2.31 ± 0.29	2.01 ± 0.06	-
Quercetin	4.88 ± 0.46	7.76 ± 0.28	8.1 ± 0.32	6.37 ± 0.46	8.76 ± 0.74	7.36 ± 0.34	8.23 ± 0.59	9.17 ± 0.42	-
IC50 (mg/mL)	18.41	17.73	17.36	17.11	20.25	24.67	22.78	25.44	3.681
R2	0.991	0.9952	0.9778	0.9834	0.9880	0.9984	0.9937	0.9925	0.9845

**Table 7 molecules-24-03052-t007:** The comparison of microwave-assisted extraction and other existing extraction methods.

Extraction Method	Extraction Time (min)	Solvent	Yield of Rutin (mg/g)	Yield of Quercetin (mg/g)	Reference
Heating reflux	180	100% MeOH	164.4–211	4.86–5.40	[28]
UAE	60	82% MeOH	208.6	−	[29]
FIASE	6	100% MeOH	202	8.033	[30]
MAE	4	65% EtOH	208.6	−	[31]
IL-PLE	5	Ionic liquid	196.3	5.18	[32]
MAE	7.5	50% MeOH	211.48	8.36	Adapted in this study

Note: UAE, FIASE, MAE, IL-PLE in the table represent ultrasonic-assisted extraction, far-infrared assisted solvent extraction, microwave-assisted extraction, and ionic liquid pressurized liquid extraction, respectively.

**Table 8 molecules-24-03052-t008:** The comparison of methods for determining the ability of chelating ferrous ions.

Samples	Chelating Agents	Solvent Volume (mL)	Reaction Time (min)	Extraction Solvent	Detection	Screening of Antioxidants	Reference
Phenolic compounds	Tetramethylmurexide	2.1	10	80% acetone	spectrophotometer	×	[37]
protein hydrolysates	Ferrozine	1.7	30	deionized water	spectrophotometer	×	[38]
Echinochrome	EDTA	−	10	96% ethanol	spectrophotometer	×	[39]
Water-Soluble polysaccharides	Ferrozine	0.7	15	96% ethanol	spectrophotometer	×	[40]
FSI(FS)	Ferrozine	0.084	5	50% methanol	Capillary electrophoresis	√	Proposed in this study

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
