# Peer review of "Microwave-Assisted Extraction Combined with In-Capillary [Fe(ferrozine)3]2+-CE-DAD to Screen Active Components with the Ability to Chelate Ferrous Ions from Flos Sophorae Immaturus (Flos Sophorae)"

_molecules, 2019, doi:10.3390/molecules24173052_

Round 1

Reviewer 1 Report

Overall, manuscript is well described. However, following changes are recommended before accepting the manuscript.

Line 105: SDS is not sodium lauryl sulfate. Please change.

Line 112: Define beta-CD here.

Line 140: Figure 1 caption is confusing. Define each optimizing parameter.  

Line 192: Table 2, if you substitute LOD or LOQ value in the regression equation, peak area(y) will be a negative value for most of analytes. This is technically not correct.  Can you explain this?

Sometimes estimating LOD or LOQ values using S/N ratio equation will not be ideal. Please provide actual LOD and LOQ values using electrophoretograms.

Reviewer 2 Report

The submitted manuscript reports a study based on a microwave-assisted extraction combined with in-capillary [Fe(ferrozine)3]2+-CE-DAD to screen antioxidant activity.

The work is well organized and written in a good english (somewhere to be revised). The analytical approach is correct and the setting up (long procedure) is described.

Results are critically discussed to show the advantages of this approach.

My opinion is for a minor revision, mainly in the grammar.

Authors should improve figure 1 that couldbe easier to read

Reviewer 3 Report

The manuscript entitled “Microwave-assisted extraction combined with in-capillary [Fe(ferrozine)3]2+-CE-DAD to screen active components with the  ability to chelate ferrous ions from Flos Sophorae Immaturus (Flos Sophorae)’ reports the [Fe(ferrozine)3]2+-CE-DAD method combined with microwave-assisted extraction to screen the activity compounds. The article is well-written and the topic is of concern for the readers of Molecules; however, there are certain weak points in the manuscript, which need to be checked and properly revised to improve its quality. Specific comments are listed below:

Comments:

P. 2: I think the length of the capillary may be an important factor for the proposed method, do authors have evaluate the effect of the capillary length? P. 9: Provide an electropherogram for all six compounds under the optimal conditions. Under these conditions, what is the electric current and how about the resolution with each compounds? P. 11: Table 5, add a column for the type of extraction solvent. P. 11: Provide the full name of the abbreviations in Table 6 (Extraction Method) as footnote. P.12: Describe more detail in how to prepare the final crude concentration of 5 mg·mL-1? Do authors evaporate the suppernatant and redissolve in extraction solvent?

Minor:

P. 7: line 182, “ug/mL-1” correct as “μg/mL-1”.

Reviewer 4 Report

The objective of the paper under evaluation was the development of a microwave-assisted extraction (MAE) method combined with an in-capillary [Fe(ferrozine)3]2+-CE-DAD method to screen for metal-chelating bioactive components from Flos Sophorae and Flos Sophorae Immaturus (FS and FSI, respectively). In this paper, the authors were able to demonstrate the ability of their developed method to screen and identify metal-chelating bioactive components, while reducing analysis time and reagent use. I recommend this study for publication, but there are some comments that I wish for the authors to consider.

Major comments:

Organisation of content The objective of the authors is completely understandable. However, the presentation and discussion of results were difficult to understand, especially when read for the first time. The authors started with method development. They aimed to optimise the separation of analytes, ferrozine and [Fe(ferrozine)3]2+. This can be confusing to the reader since the readers perceive that the authors have yet to determine the components of FS and FSI. It is only upon further reading that it was clear that the components of FS and FSI are established. Perhaps the authors may either inform the reader beforehand of the components of FS and FSI so that the method development section is clearer for the reader. In the title, the authors placed “microwave-assisted extraction” first followed by “in-capillary [Fe(ferrozine)3]2+-CE-DAD”. To the reader’s mind, it is expected that the authors will discuss MAE first followed by method development. Therefore, I suggest the authors to discuss the results of MAE first over the optimisation of the on-line method. This flow is not only consistent with your title, but also has a better discussion flow, since this will bring the method development section closer to the method validation and method application sections. I do not think it is necessary to discuss in length the comparison of the authors’ developed methods with existing methods. They only need embed it in the main discussion of method development and MAE parameter optimisation. Therefore, I suggest to remove Section 2.5 altogether and move the necessary discussions in the other relevant sections. Place Figure 3 right after Section 2.4.1 (Screening of antioxidants of FSI(FS)). In Section 3.4 (The condition and principle of method), the authors described the optimised CE running parameters and optimised BGS formulation. However, this is not a true representation of what transpired in the authors’ laboratory. I suggest that the authors describe instead which parameters they optimised.

Use of β-CD as background solution (BGS) additive

The authors mentioned the use of β-CD as β-CD additive. In their discussion (lines 113-114), they mentioned that “β-CD has an important influence on the migration and separation of ligands in electrophoresis especially in separating chiral compounds”. While this may be of value in future applications where the analytes may be chiral in nature, I question the authors’ motive for choosing to add β-CD as BGS additive in this study. The analytes mentioned in this paper do not have chiral centres in their aglycone moieties. Unless their purpose for adding β-CD in the BGS is to modify selectivity, the authors have to justify its use for screening for metal-chelating bioactive components of FS and FSI.

Also, the reference they used to justify the use of β-CD in their BGS (reference 31) points to β-CD as a chiral selector in capillary electrophoresis. If the authors do not wish to use β-CD as chiral selector, then they should at least not use this reference.

References

Relevent review articles such as those shown below should be referenced and discussed to show the state of the art:

Wuethrich, A., Quirino, J.P. Derivatisation for separation and detection in capillary electrophoresis (2015–2017). (2018) Electrophoresis, 39 (1), pp. 82-96.

Wang, W.-F., Yang, J.-L. Advances in screening enzyme inhibitors by capillary electrophoresis. (2019) Electrophoresis, DOI: 10.1002/elps.201900013 Ramos-Payán, M., Ocaña-Gonzalez, J.A., Fernández-Torres, R.M., Llobera, A., Bello-López, M.Á. Recent trends in capillary electrophoresis for complex samples analysis: A review. (2018) Electrophoresis, 39 (1), pp. 111-125.  Kašička, V. Recent developments in capillary and microchip electroseparations of peptides (2015–mid 2017). (2018) Electrophoresis, 39 (1), pp. 209-234.

 Minor comments

Please define your graphs in Figures 1 and 2 in the figure captions. Figure captions must be stand alone and should not make the reader refer to the text. Please label the analyte signals in Figure 3. It would be difficult for the reader to make them refer to a separate Figure 4. Having said this, I believe that a separate Figure 4 is no longer needed. Choose whether to use M or mol L-1. Otherwise, there is no need to define these units. Please check the consistency in the spelling of ferrozine. Other minor comments:

Lines 50-51: I suggest to replace “have no” with “does not have”

Line 53: “possessed” should read as “possesses”

Lines 53-54: Please clarify what the authors meant by “complicated operation”. Is it the operation of UV/Vis spectrophotometers or the metal chelating assays?

Line 61: Please italicise “Sophora japonica”.

Lines 69-72: Please break into simpler sentences. Also, please define the meaning of MAE in line 71.

Line 120: “focused” should read as “focusing”

Equation [a]. “inhibitor rate” should read as “inhibition rate”

Line 199 and Figure 3 caption: I don’t think “mixture” is the right word

Lines 199-202. Please break into smaller sentences.

Lines 295-296 and line 305. Please rephrase.

Line 331. “sensitive” should read as “sensitivity”

Round 2

Reviewer 1 Report

Line 193: Please update LOD and LOQ range with new values.

Reviewer 4 Report

The authors have considered all the comments of this reviewer and made appropriate changes to the revised manuscript.

Author Response

Thank you sincerely for your meaningful comments